# Genus *Lepisanthes*: Unravelling Its Botany, Traditional Uses, Phytochemistry, and Pharmacological Properties

**DOI:** 10.3390/ph15101261

**Published:** 2022-10-13

**Authors:** Nadia Mohamed Tarmizi, Syarifah Aisyah Syed Abd Halim, Zubaidah Hasain, Elvy Suhana Mohd Ramli, Mohd Amir Kamaruzzaman

**Affiliations:** 1Department of Anatomy, Faculty of Medicine, Universiti Kebangsaan Malaysia, Jalan Yaacob Latiff, Cheras, Kuala Lumpur 56000, Malaysia; 2Preclinical Department, Faculty of Medicine & Defence Health, Universiti Pertahanan Nasional Malaysia, Sungai Besi, Kuala Lumpur 57000, Malaysia

**Keywords:** *Lepisanthes*, Sapindaceae, natural product, traditional medicine, pharmacological, antioxidant, antimicrobial, antihyperglycemic, antidiarrheal

## Abstract

Extensive knowledge related to medicinal characteristics of plants by living in forest or semi-forest habitats and close observations of indigenous communities have led to the discoveries of the genus *Lepisanthes* and its traditional uses. The genus *Lepisanthes* is a member of the Sapindaceae family and is found in various regions of the world. Six species of *Lepisanthes* such as *L. alata*, *L. amoena*, *L. fruticosa*, *L. senegalensis*, *L. rubiginosa*, and *L. tetraphylla* are widely utilized in traditional and folk medicinal systems. They have been used for centuries for the treatment of ailments or symptoms such as pain, dizziness, high fever, frequent passing of watery stool (diarrhea), abscess, and healing of cuts and wounds. Various methodological approaches, mainly in vitro studies, have been employed to further explore the roles of the genus *Lepisanthes*. The studies identified that the genus *Lepisanthes* exerts beneficial effects such as antioxidant, antimicrobial, antihyperglycemic, antimalarial, analgesic, and antidiarrheal. However, the summary of the available literature remains inconclusive. This review aims to comprehensively address the botany, traditional uses, phytochemistry, methods, and pharmacological properties of the six commonly used *Lepisanthes* species. Hence, our review provides a scientific consensus that may be essential in translating the pharmacological properties of the genus *Lepisanthes* into future novel cost-effective medicines.

## 1. Introduction

People in developing countries around the world are dying every day from preventable diseases due to low levels of education, limited basic health facilities, and uncertain economic status [1]. Those who manage to survive from diseases rarely recover completely as they are mostly malnourished. Therefore, preventive approaches should consider socioeconomic aspects, particularly in developing countries [1]. Traditional medicine is an affordable alternative approach to modern medicine to prevent diseases. The World Health Organization (WHO) Expert Group defined traditional medicine as the total of all knowledge and practices, whether explicable or not, used in the diagnosis, prevention, and elimination of physical, mental, or social imbalances that rely solely on practical experience and observation passed down verbally or in form of writing from generation to generation [2]. Medicinal plants can contribute to illness prevention initiatives in countries such as African countries and some regions in Asia, given the vast biodiversity of plants in the regions and the relatively cheaper cost of using plant-derived medications rather than processed synthetic treatments.

Plants can produce a wide range of bioactive compounds. Fruits and vegetables store prominent levels of phytochemicals, which may protect against free radical damage. Plants that contain beneficial phytochemicals can help the human body meet its antioxidant requirements by functioning as natural antioxidants. Antioxidants are abundant in many plants. Vitamins A, C, and E, as well as phenolic chemicals (i.e., flavonoids, tannins, and lignins) found in plants act as antioxidants [3]. Currently, the focus on the utilization of medicinal plants had been on disease prevention rather than disease treatment alone. The reports in the literature from current research on the use of medicinal plants and their ingredients in illness prevention are substantial. Among the medicinal plants, *Lepisanthes* is a genus that originated from tropical countries around the globe. The name of the genus *Lepisanthes* is taken from a Latin word describing the flower which has scales on the inner surface of the petals. *Lepis* means scale, and *anthos* refers to flowers. Most of the species are comprised of trees and shrubs [4]. In recent years there has been rising interest to explore the constituents of the *Lepisanthes* under distinct species, and the studies were gradually extended to animal experiments to assess the potential of pharmacological properties that could be regarded as a medicinal herb. The six *Lepisanthes* species that are commonly investigated are *L. alata*, *L. amoena, L. fruticosa, L. senegalensis, L. rubiginosa,* and *L. tetraphylla*. Therefore, in this paper, we aim to review the botany, traditional uses, phytochemistry, methodological approaches, and pharmacological effects of six *Lepisanthes* species.

## 2. The Botany of Genus *Lepisanthes*

The genus *Lepisanthes* consists of 26 species [4]. The geographical distribution of *Lepisanthes* is broad, extending from India and Sri Lanka in the west to Malaysia, the Philippines, Vietnam, and Papua New Guinea in the east. In Southeast Asia, six of these species are native to Malaysia and can be found flourishing in the wild as well as in orchards (i.e., *L. alata*, *L. amoena, L. fruticosa, L. senegalensis, L. rubiginosa,* and *L. tetraphylla*) [5,6]. The tropical rainforest is the natural habitat of *Lepisanthes* species, which grows from lowland to lower montane forest at 1800 m above sea level. It can be also found in streams, wastelands, coastal forests, islands, or in the transition areas between mangroves and dry land. It grows in the organic loam, with the addition of coarse sand or perlite to improve its drainage. Some of these species are still found in the wild, while others are cultivated as ornamental trees and conserved for their delicious fruits [6]. *Lepisanthes* is found widely distributed in the aforementioned geographical areas as it grows in a humid climate with year-round rainfall, temperatures between 25 and 28 °C, and a distinct dry season [7]. It is a shrub tree of about 4–10 m in height. The leaves are imparipinnate with a pair of leaflets and a single terminal leaflet at the terminal end. Its fruits are in bunches like grapes and are shiny red to purple in color. The ripe fruit turns dark purple and contains approximately one to four ellipsoid seeds with fleshy aril. The color of the fruits attracts birds, which help to disseminate the seeds and increase their reproducibility [8]. Figure 1 shows images of *L. alata*, also known as Ceri Terengganu in Malaysia.

## 3. Traditional Uses of Genus Lepisanthes

Traditional medicine is an ancient and culturally based healthcare approach that differs from scientific medicine, and it is passed down by many civilizations [9]. In the past, it was used either as a mixture of various ingredients or as concentrated plant extract without isolation of the active compounds [10]. Traditional medicine has contributed enormously to humanity in the fight against diseases and to living a healthy and long life.

According to ethnobotanical studies, certain *Lepisanthes* species are commonly consumed as food sources by native people and are also applied in traditional medicine [11]. As seen in Table 1, its usage is dependent on the type of species and parts of the plant. For instance, *Lepisanthes tetraphylla* in Malaysia is used in the treatment of cough and fever [12]. In the South Asian region, the roots of *L. tetraphylla* are mixed with other herbal plants and are consumed to treat diarrhea in Bangladesh [13], and its seed has been included in medicated shampoo to resolve dandruff problems among Indians in Tamil Nadu [14]. In Thailand, the roots of *L. senegalensis* are used to treat malaria, fever with vertigo, chest pain, and nosebleed [15]. The roots are also used to cure diarrhea in Bangladesh [16]. The leaves of *L. senegalensis* is used in Senegal to treat bacterial and fungal infections, pain, inflammation, and asthenia [17]. Interestingly, it has been reported that it is also used in treating lung infections, including pleuropneumonia in farm animals [18]. On the other hand, the roots of *L. fruticosa* have been used as a paste to reduce itchiness and lower body temperature during fever in traditional Malay medicine [19,20]. The Kedayan in Sarawak regularly consumes decoction tea made from the roots of *L. fruticosa* to prevent rheumatism and backache and to maintain vitality [20]. In Borneo, the leaves of *L. amoena* are known to be used as part of traditional cosmetic ingredients, for example, in face powder, skin cleanser, soap, and shampoo [21]. Furthermore, in Indonesia, the Dayak tribe of East Kalimantan utilizes the leaves of *L. amoena* to treat facial skin problems by scrubbing the young leaves until they appear foamy before applying them to the skin [22]. Traditionally, in Terengganu, Malaysia, the young leaf of *L. rubiginosa* is crushed and applied to reduce muscle soreness, while the ripe fruits are eaten to reduce fever, flatulence, and postpartum blues [23]. In addition, *L. rubiginosa’s* fruit is consumed to treat diarrhea, dysentery, and jaundice by local people in the Barisal district, Bangladesh [24]. In East Kalimantan, the leaves of *L. alata* are used to reduce skin itchiness due to scabs [25].

## 4. Phytochemistry

Phytochemicals are active chemical compounds biologically found in plants that contribute health benefits for humankind. They are classified into primary and secondary metabolites. Primary metabolites are responsible for the biochemical activity of the plant (i.e., photosynthesis and respiration), while secondary metabolites are chemical compounds that are important in protecting the plant against damage [26]. Flavonoids, tannins, terpenes, phenolic compounds, and glycosides are the main secondary metabolites that have been isolated and investigated in several *Lepisanthes* species [4]. In this review, we listed 36 compounds that are related to our pharmacological activity discussion (Table 2).

Salahuddin et al. [19] identified two main groups/classes of phenolic compounds, namely, flavonoids and tannins, from an ethanolic seed crude extract of *L. fruticosa*. A total of six subclasses (i.e., flavanol, flavanonol, flavonol, flavone, isoflavone, and anthocyanin) of flavonoids class that consists of 17 compounds was found to be present in *L. fruticosa* (Table 2). Moreover, a total of five compounds of the tannin class and three other compounds was also found in *L. fruticosa* (Table 2). Their fraction as the bioactive compound could be contributing to their outstanding antioxidant and α-glucosidase inhibition activity [19].

Lupane and hopane, which are derivatives of pentacyclic triterpenes, were isolated from the stem and roots of *L*. *senegalensis* [15]. The 3-*O*-*trans*-caffeoylbetulinic acid compound was among the main lupane compounds that was isolated from *L*. *senegalensis* and was proven to have moderate antimalarial activity [15] (Figure 2). In addition, the proanthocyanidines of mature *L. alata* leaves showed a high degree of polymerization composed of (epi)gallocatechins (Figure 3) and (epi)catechins (Figure 4), compounds linked through B-type 4-8 interflavanyl bonds. These compound might be responsible for the antihyperglycemic effect [27].

## 5. Pharmacological Activity

Table 3 summarizes the pharmacological activities of the *Lepisanthes* species. The species that are commonly investigated pharmacologically are *L. alata* (Blume) Leenh, *L. amoena* (Hassk.) Leenh, *L. fruticosa* (Roxb.) Leenh, *L. senegalensis* (Poir.) Leenh, *L. rubiginosa* (Roxb.) Leenh, and *L. tetraphylla* (Vahl.) Radlk. Most of the studies were performed in vitro except for Hasan et al. [28]. The genus *Lepisanthes* possesses specific pharmacological properties (i.e., antioxidant activity, antimicrobial activity, antihyperglycemic, analgesic, and antidiarrheal) depending on the species and parts of the plants. Based on the studies, the antioxidant activity showed the most promising pharmacological properties for the genus *Lepisanthes*. Nevertheless, the safety and efficacy of these genera have been suboptimally investigated [15,17,29]. The subsequent subheadings will discuss the pharmacological properties of the six *Lepisanthes* species in detail.

### 5.1. Antioxidant Activities

The antioxidant activity of a plant is considered one of the most important pharmacological properties. Many in vitro methods have been designed for screening the antioxidant activities in plants [36]. The two most common methods to test for antioxidants in vitro are free radical scavenging methods: 2,2-diphenyl-1-picrylhydrazyl (DPPH) and 2,2-azinobis (3-ethyl benzothiazoline-6-sulfonic acid) diammonium salt (ABTS). Both assays are based on electron transfer reactions. The DPPH assay is the most used, as this method is straightforward [28]. It is based on the reduction of a stable free radical which is a purple color that turns yellow upon reacting with antioxidants with maximum absorption at 517 nm. The ABTS assay, however, is reported to be more diverse, as it can determine the antioxidant capacity of both hydrophilic and lipophilic plant extracts [37]. The additional method to screen antioxidant activity is the beta-carotene bleaching assay. In this assay, linoleic acid oxidation will form free radicals that oxidize β-carotene. The presence of antioxidants from the plant will neutralize the free radicals and slow down the rate of β-carotene bleaching [19].

Looi et al. [5] compared the in vitro antioxidant activity of 60% ethanolic extract of *L. alata*’s peel, flesh, and seed using both DPPH and ABTS assays. Butylated hydroxytoluene (BHT), vitamin E, and vitamin C were selected as the positive controls for DPPH methods, while Trolox was the positive control for the ABTS method. This study proved that 60% ethanolic extract of *L.*
*alata* seed showed the highest antioxidant capacity (83.9%) compared to the peel (83.2%) and flesh (52.4%) using DPPH free radical scavenging methods. In addition, the antioxidant activities of seed and peel were significantly higher than the BHT (58.2%) and vitamin E (42.7%) but were comparable to vitamin C (88.2%). The seed (48.2%) and peel (45.1%) extracts consistently showed significantly higher antioxidant activity compared to the flesh (33.9%) extract using the ABTS assay test, despite the percentage of the antioxidant activity being lower compared to the DPPH test. However, the antioxidant activities of seed and peel were significantly lower than the positive control Trolox (64.7%). At the end of the study, it was concluded that the high antioxidant activity in seed and peel extracts might be due to the high content of both total phenolics and total flavonoids [5]. Anggraini et al. [29] investigated the DPPH radical scavenging activities of three different solvents with different polarities (i.e., aqueous, methanol, and ethanol) in different parts of *L. alata* (i.e., rind, flesh, seeds, whole fruits, leaves, and bark). The results showed that ethanol and methanol extracts have higher DPPH radical scavenging activities compared to the aqueous extract except for flesh. The ethanolic extract from the bark showed the highest antioxidant activity (93%) followed by the seeds (90%), rind (86%), leaves (80%), whole fruit (46%), and flesh (21%) [29].

Salahuddin et al. [19] evaluated the in vitro antioxidant activities of pulp and seed of unripe *L. fruticosa* fruit using various solvents: hexane, chloroform, ethyl acetate, and ethanol. The authors selected the unripe fruit because they had confirmed in a previous study that the antioxidant activity of *L. fruticosa* fruit had reached the highest levels at the unripe stage compared to eight different maturity stages [38]. In the present study, the DPPH test showed that the ethanolic extract of both seed and pulp had higher antioxidant activity followed by ethyl acetate extract. The lowest antioxidant activity was recorded by pulp and seed hexane extract. Furthermore, the ethanolic crude seed extract of *L. fruticosa* showed significantly stronger scavenging activity (IC_50_ 0.178 mg/mL) as compared to the pulp (IC_50_ 0.207 mg/mL) [19]. It was also found that the ethanolic crude seed extract had a powerful antioxidant capacity compared to the reference standards (BHT (IC_50_ 1.154 mg/mL), vitamin C (IC_50_ 0.087 mg/mL), and vitamin E (IC_50_ 0.210 mg/mL)) [19]. Apart from the DPPH test, the extracts were also tested for β-carotene bleaching activity. The ethanolic seed extract revealed the significantly highest antioxidant activity compared to positive control vitamin C (70.7% vs. 25.8%). Overall, this study highlighted that *L. fruticosa* ethanolic crude seed extract had the highest antioxidant activity using DPPH and β-carotene bleaching assays. Furthermore, ethanol, with its high polarity has been proven to be effective in extracting the antioxidant compound, especially in the crude seed extract of *L. fruticosa*, compared to other solvents used [19].

Hasan et al. [28] studied the antioxidant activity of *L. rubiginosa* ethanolic leaf extracts. They reported that the DPPH test of ethanolic leaf extracts had a relatively good antioxidant activity (IC_50_ 31.62 μg/mL) compared to the standard ascorbic acid (IC_50_ 12.02 μg/mL). The antioxidant activity in this extract is possibly due to its phenolic content (422.42 mg gallic acid equivalent (GAE)/100 g of dried extract). This signifies the presence of the hydroxyl group, which is responsible for scavenging the free radicals [28]. Salusu et al. [21] tested the antioxidant activity of an ethanolic fruit extract of *L.*
*amoena*, which was later divided into flesh, seed, and pericarp [21]. Based on the antioxidant results, it was then categorized as strong (<50 ppm), active (50–100 ppm), moderate (100–250 ppm), weak (250–500 ppm), or inactive (>500 ppm), as described by Jun et al. [39]. The authors identified the parts of the genus with the highest antioxidant activity as pericarp, seed, and flesh (IC_50_ values of 53.21 ppm, 63.31 ppm, and 122.51 ppm, respectively). Salusu et al. [21] concluded that the ethanolic pericarp and seed extracts of *L. amoena* had potential as natural sources of antioxidants, which might be due to the presence of flavonoid contents. However, the ascorbic acid that was used as a positive control showed stronger antioxidant activity (IC_50_ value 3.06 ppm) compared to the rest [21]. Batubara et al. [31] also reported the antioxidant activities of the stem and leaves of *L. amoena*, which were extracted with methanol and 50% ethanol. The results showed that methanolic stem and leaf extracts exhibited IC_50_ values of 99.10 µg/mL and 17.25 µg/mL, respectively, while 50% ethanolic stem and leaf extracts exhibited IC_50_ values of 50.0 µg/mL and 9.76 µg/mL, respectively. The positive control (catechin) exhibited an IC_50_ value of 2.94 µg/mL. Therefore, all the extracts were noted to be unable to inhibit the oxidation reaction of DPPH by 50% [31]. Table 4 summarizes the above-mentioned antioxidant activities.

### 5.2. Antimicrobial Activities

Continuous studies are being carried out to find an alternative treatment for antimicrobial infections by using extracts from herbal plants due to increasing numbers of antibiotic resistant microorganisms. *Lepisanthes* species were reported to exhibit antimicrobial activities, especially antibacterial and antifungal activities against the most common microorganisms. Purnamasari et al. [32] utilized the agar diffusion method and n-hexane, ethyl acetate, and ethanol solvents to compare the antimicrobial activity of different maturity (young, semi-mature, and mature) leaves of *L. amoena* against common microorganisms that can cause acne, which are two Gram-positive bacterias (*Propionibacterium acnes*, *Streptococcus mutans*), and one fungus (*Candida albicans*). The results showed that the mature leaves’ antimicrobial activity was stronger, specifically in the ethanolic extract. They exhibited the highest inhibition zone against *P. acnes* (12.00 ± 0.00 mm) and *C. albicans* (16.11 ± 0.19 mm). At the same time, this study also estimated the total phenolic content in the *L. amoena* leaf extracts using the Folin–Ciocalteu reagent. The mature leaves that were extracted using ethanol as solvent showed the highest phenolic content (0.87 mg GAE/g dry weight) compared to the rest. Aligned with the previous study, the authors observed that the antimicrobial property has a positive linear correlation with the total phenolic content of *L. amoena* leaves, strengthening the link between phenolic acids, the phenolic moiety, and organic antimicrobial activity [32]. An antibacterial study was performed by Batubara et al. [31] using an antibacterial assay on methanolic and 50% ethanolic *L. amoena* stem and leaf extracts against Gram-positive *Propionibacterium acnes*. Only the stem for both methanol and 50% ethanol extracts showed a minimum inhibitory concentration (MIC) value of 1.0 mg/mL, and only the 50% ethanolic stem extract showed a minimum bactericidal concentration (MBC) value of 2.0 (mg/mL). However, these results indicated that both extracts were not effective as antibacterial agents compared to the controls used, which were chloramphenicol, tetracycline, and isopropyl methylphenol [31].

Barua et al. [33] evaluated the antimicrobial activity of methanolic bark extract of *L. rubiginosa* against both Gram-positive (i.e., *Staphylococcus aureus* and *Bacillus cereus*) and Gram-negative bacteria (i.e., *Salmonella typhi* and *Shigella dysenteriae*). The authors used the disc diffusion method with cephradin (first group cephalosporin) as a control. The study revealed that a significant zone of inhibition was present in both Gram-negative bacteria but only in one Gram-positive bacteria (*S.*
*aureus*). The maximum resistance was shown by Gram-positive *B. cereus*. It was postulated that Gram-negative bacteria were more susceptible to the extract perhaps due to the presence of the broad-spectrum antibacterial compound in the bark. Moreover, the presence of the phenolic compound, saponins, alkaloids, and flavonoids identified by qualitative tests might contribute to antimicrobial activity in this study [33]. Additionally, the extract was also tested for antifungal activity against *Aspergillus niger* and *C.*
*albicans*, but the extract had minimal antifungal activity compared to standard griseofulvin.

Dior et al. [17] reported that the ethanolic leaf extract of *L. senegalensis* showed greater antimicrobial activity against bacteria as compared to fungi [17]. In this study, two Gram-positive bacteria, *S. aureus* and *Enterococcus faecalis*, and two Gram-negative bacteria, *Pseudomonas aeruginosa* and *Escherichia coli*, were tested. The antibacterial activity of the extract was higher than the positive control, gentamicin, against both Gram-positive bacteria. The extract, however, showed that the antimicrobial activity on *E. coli* was equal to gentamicin but was not effective against *P. aeruginosa*. Moreover, the extract exhibited much lower antifungal activity compared to positive control amphotericin B in all three tested fungi: *Aspergillus fumigatus*, *C. albicans*, and *Cryptococcus neoformans*.

In addition, Looi et al. [5] tested a 60% ethanolic extract of *L. alata’s* seed, peel, and flesh against four Gram-positive bacteria (i.e., *Bacillus subtilis*, *B. cereus*, *Listeria monocytogene*, and *S. aureus*) and three Gram-negative bacteria (i.e., *Salmonella enterica* serovar Typhimurium, *E. coli*, and *P. aeruginosa*). Overall, the ethanolic seed extract showed the highest antimicrobial activity towards most Gram-positive bacteria except for *L. monocytogenes*. Even though *L. monocytogens* is a Gram-positive bacteria, it can form a biofilm to protect against antimicrobial agents [5]. Moreover, all parts of *L. alata* failed to show an inhibitory effect against all tested Gram-negative bacteria, possibly due to the presence of lipopolysaccharides within the wall of Gram-negative bacteria, which protects the bacteria from antimicrobial agents [40].

Nowadays, silver nanoparticles synthesized from plants are currently in high demand due to their vast application in pharmaceuticals. Meena et al. [35] proved that silver nanoparticles (AgNPS) synthesized using an aqueous leaf extract of *L. tetraphylla* at 0.03–0.05 mg concentration showed significant inhibition of bacterial growth against many clinical strains, methicillin-resistant *Staphylococcus aureus* (MRSA), extended-spectrum beta-lactamase-producing *E.*
*coli* (ESBL *E. coli*), multidrug-resistant *P.*
*aeruginosa*, and multidrug-resistant Acinetobacter species. In comparison, the antimicrobial activity of the crude methanolic *L. tetraphylla* leaf extract without nanoparticles (AgNPs) was just able to inhibit the growth of ESBL *E. coli* only at a higher concentration (1 mg/0.1 mL). This finding is important, as AgNPs synthesized using *L. tetraphylla* leaf extract showed broad-spectrum antimicrobial activity and should be explored further to be incorporated in dressing for wounds or burn patients [35]. Table 5 summarizes the antimicrobial activities of genus *Lepisanthes*.

### 5.3. Antihyperglycemic Activities

Diabetes mellitus, especially Type II (T2DM), is a complex chronic disease that affects worldwide populations. Serious complications such as diabetic nephropathy, retinopathy, and neuropathy remain common health issues, as 90% out of 422 million people worldwide are diagnosed with T2DM [41]. Usually, the control of postprandial hyperglycemia in adults with T2DM is challenging. The α-amylase enzyme released by the pancreas and salivary glands is the key enzyme that breaks down dietary carbohydrates into simpler monosaccharides [19], while the α-glucosidase enzyme is responsible for degrading it further to glucose, which will enter the bloodstream upon absorption. Theoretically, postprandial hyperglycemia could be alleviated by reducing the production and absorption of glucose through inhibition of these carbohydrate hydrolyzing enzymes (i.e., α-amylase and α-glucosidase enzymes) [41]. Nonetheless, most oral antihyperglycemic drugs (i.e., acarbose, miglitol, and voglibose) have unpleasant gastrointestinal side effects including bloating, diarrhea, and abdominal discomfort. Alternatively, vast research on the source of edible medicinal plants with potential use to inhibit α-amylase and α-glucosidase, and isolation of their active compounds, would be ideal to prevent side effects from oral antihyperglycemic drugs. The antihyperglycemic assay is usually performed by using α-glucosidase inhibition and α-amylase inhibition assays [27]. Traditionally, plants that are used as antihyperglycemic agents exert a higher α-glucosidase inhibitor effect compared to the α-amylase [19]. To date, limited studies have explored the antihyperglycemic activities of the genus *Lepisanthes*. Two species, *L. fruticosa* and *L. alata*, were evaluated using in vitro techniques, while *L. rubiginosa* was assessed using in vivo techniques.

Salahuddin et al. [19] determined the antihyperglycemic activity using the seed and pulp of *L. fruticosa* with various solvents; hexane, chloroform, ethyl acetate, and ethanol. They reported that the ethanolic seed crude extracts of *L. fruticosa* and its fraction (M4), exert significantly the highest inhibitory effect (IC_50_ 1.873 μg/mL) compared to acarbose. The ethanolic extract was found to have a high amount of phenolics (120.204 mg GAE/g), whereby flavonoids and tannins were the major phenolics that may be responsible for the antihyperglycemic effect [19]. The potential use of *L. alata* leaves as prevention for postprandial hyperglycemia was investigated in vitro by Zhang et al. [27]. Overall, this study showed that the mature aqueous leaves extract of *L. alata* exhibited inhibitory activities of 13.20 mmoL AE/g (acarbose equivalent/g) for α-amylase, and 0.3 mmoL AE/g for α-glucosidase while the proanthocyanidins of mature leaves showed higher potential of 17.8 mmol AE/g for α-amylase and 0.5 mmol AE/g for α-glucosidase. This indicates that the proanthocyanidins of mature leaves extract are 8.5 times more powerful than acarbose in inhibiting α-amylase. They also identified that the bioactive compounds that are possibly responsible for the antihyperglycemic effect are the proanthocyanidins, composed of (epi) catechins and (epi) gallocatechins, linked through B-type 4-8 interflavanyl bonds [27]. Further study was done in comparing the antihyperglycemic effect between aqueous extract and proanthocyanidins of young versus mature leaves. The results showed that the aqueous extract and proanthocyanidins of young leaves are more potent than acarbose by only 2.6 and 5.3 times in inhibiting α-amylase compared to α-glucosidase. Both results showed that the proanthocyanidins of both young and mature leaves have the potential to be used as selective α-amylase in the treatment of diabetes. They hypothesized that the higher α-amylase inhibition of the mature leaf might be due to the higher mean degree of polimerization of proanthocyanidins which exert better inhibitory activities against starch hydrolase [30].

An in vivo study by Hasan et al. [28] revealed that an ethanolic leaf extract of *L. rubiginosa* showed an antihyperglycemic effect in Swiss-albino mice [28]. A total of 20 mice was divided into four groups. The control group received 1% Tween 80 and water, the standard group received 5 mg/kg of glibenclamide, and the two intervention groups received 250 mg/kg and 500 mg/kg of the ethanolic leaf extract, respectively. The blood was collected from the tail vein to monitor oral glucose tolerance. Both intervention groups (250 mg/kg and 500 mg/kg) showed a significant reduction in blood glucose levels compared to the control group. The authors postulated *L. rubiginosa* might exerts antihyperglycemic activity by either promoting pancreatic insulin secretion or increasing glucose uptake in tissues [28]. Further investigation needs to be carried out to ascertain the actual mechanism of antihyperglycemic activity in this study. Table 6 summarizes the antihyperglycemic activities of the genus *Lepisanthes*.

### 5.4. Antidiarrheal Activities

The antidiarrheal effect of ethanolic extract of *L. rubiginosa* leaves was studied in the castor oil-induced diarrhea model in experimental Swiss-albino mice (Table 7). Four groups were designed, and each group consisted of five mice. The control group received 1% Tween-80 with water, the standard group received loperamide 3 mg/kg, while the two intervention groups received extract doses of 250 mg/kg and 500 mg/kg body weight. After 60 min of intervention, all the mice were given 0.5 mL of castor oil via oral gavage to induce diarrhea. The frequency of defecation was observed for four hours and counted by placing a white blotting paper in the individual cage. The papers were changed every hour for a total of four hours. Then, the percentage of inhibition of defecation was calculated. The result showed that the highest percentage of inhibition of defecation was in the standard loperamide group (88.59%) followed by the intervention group, which was given 500 mg/kg of extract (77.19%), and the last was the other intervention group, which was given 250 mg/kg of extract (57.89%). The findings suggested that the ethanolic leaf extract of *L. rubiginosa* exerts an antidiarrheal effect in a dose-dependent manner [28]. Nevertheless, the identification of specific bioactive compound that may responsible for the antidiarrheal effect is still uncertain.

### 5.5. Analgesic Activities

Hasan et al. [28] also demonstrated the analgesic effect of ethanolic leaf extract of *L. rubiginosa*. A total of 20 Swiss-albino mice was divided into four groups (Table 7). The control group received 1% Tween-80 and water, the standard group received Diclofenac-NA 25 mg/kg, and the two intervention groups received extract doses of 250 mg/kg and 500 mg/kg body weight, respectively, via oral gavage. After 30 min of administrating oral gavage treatment according to groups, 0.7% *v*/*v* acetic acid solution was injected intraperitoneally into all mice to induce pain. The mice were observed for 15 min at 5 min intervals, and the frequency of writhing was measured. The results were then calculated as the percentage of inhibition. In this study, the highest percentage of inhibition of the writhing reflex was noted in the standard group (86.52%), the second highest was in the intervention group, which received 500 mg/kg of extract (58.43%), and the lowest was in the intervention group that received 250 mg/kg of extract (46.07%). Although the percentage of inhibition was lower compared to the standard drug tested, the results confirmed the dose-dependent manner of the ethanolic leaf extract’s analgesic activities [28].

### 5.6. Antimalarial Activities

Lomchid et al. [15] elucidated 7 out of 12 isolated compounds from ethyl acetate *L. senegalensis* stem and root extracts for antimalarial activity. By identifying and comparing the spectrophotometric data with the literature data, compound no. 6 was identified as 3-*O*-*trans*-caffeoylbetulinic acid. An in vitro study was conducted using cultured *Plasmodium falciparum* in human erythrocytes, while the quantitative assessment of antimalarial activity was carried out by the microculture radioisotope technique. Compound no. 6 was also compared with the standard compounds (mefloquine and dihydroartemisinin). The antimalarial activity exhibited by mefloquine and dihydroartemisinin was IC_50_ 0.029 and 0.0024 µM, respectively. The results showed that the bioactive compound no. 6 exhibited moderate antimalarial activity against *P. falciparum* with an IC_50_ value of 4.5 µM [15].

Widyawaruyanti et al. [34] evaluated in vitro antimalarial screening of 20 leaves and stems of the plant collected from exploration in Alas Purwo National Park, East Java, Indonesia (Table 7). One of the plants was identified as *L. rubiginosa* and was extracted with 80% ethanol by the ultrasonic assisted maceration technique. The 20 extracts were tested for in vitro antimalarial activity against the *P. falciparum* 3D7 strain (chloroquine-sensitive) using the histidine-rich protein II (HRP2) assay. The HRP2 antimalarial assay was performed using ELISA. Among the extracts, the 80% ethanolic stem extract of *L. rubiginosa* showed the highest inhibition value (92.4%) at a concentration of 1000 µg/mL. However, *L. rubiginosa’s* IC_50_ value (IC_50_ 252 µg/mL) was categorized as inactive antimalarial according to the in vitro, antimalarial activity criteria of Chinchilla et al. [42]. Table 7 summarizes the methodological approaches and the rest of the pharmacological activities of the genus *Lepisanthes*.

## 6. Toxicity

At present, although few *Lepisanthes* species are used in traditional medicine, reports on its safety are still scarce. Anggraini et al. [29] studied the toxicity of the whole fruit, leaves, and bark of *L. alata* using the brine shrimp lethality bioassay test. The concept of the bioassay was to place different concentrations of sample extract (10, 100, and 1000 ppm) in a vial containing 5 mL of seawater with ten shrimps (larva *Artemia salina* Leac) [29]. At the same time, a blank control group (without extract) was also tested. After 24 h, the number of dead shrimps and the lethality concentration (LC_50_ dose that was lethal for half the shrimps) were calculated by probit analysis. The LC_50_ value of <1000 mg/mL is considered toxic. In this study, the results showed that the lowest LC_50_ was 1600 mg/mL for the whole fruit extract, which indicated that none of the tested parts were toxic as the values were > 1000 mg/mL regardless of their concentrations.

Lomchid et al. [15] investigated a cytotoxicity assay on a few compounds from the ethanolic stem and root extract of *L. senegalensis*. This assay was performed against the human epidermoid carcinoma in the mouth (KB) and human small cell lung cancer (NCI-H187) using microplate assay (REMA), as well as cytotoxicity test against primate cell line (Vero) using the green fluorescent protein (GFP) detection method. Results showed that Triterpenes 1 and 4–6 exhibited cytotoxicity against the NCI-H187 cell line with IC_50_ values of 31.5, 28.5, 16.2, and 4.0 µM, respectively. However, these compounds also showed cytotoxicity against Vero cells, with IC_50_ values of 75.5, 16.6, 8.9, and 5.0 µM, respectively [15]. In addition, Dior et al. [17] performed a cytotoxicity test of *L. senegalensis* ethanolic leaf extract using tetrazolium-based colorimetric assay (MTT assay), against Vero cells with Berberine used as a positive control. The results showed that the extract was more toxic than the positive control [17]. In this case, further evaluation and isolation of the active compound should be made if this plant extract is to be given orally as the high toxicity might be attributed to the synergistic effects of other compounds existing in the plant extract. Overall the finding from these studies [15,17] may be contradicting Anggraini et. al [29] due to the different species and methods used to estimate the safety of the genus *Lepisanthes* [17]. 

## 7. Future Perspective

Even though *Lepisanthes* species are widely distributed in Southeast Asian nations and have been historically utilized by the locals to treat diseases, we know relatively little about their advantages in terms of scientific research findings. Previous studies have indicated that *Lepisanthes* possesses several pharmacological activities such as strong antioxidant, antimicrobial, antihyperglycemic, analgesic, and antidiarrheal effects that are fundamental to various therapeutic purposes. However, we observed variations in terms of study design and methods among the studies, such as solvents used for extraction, control, dose, type of species, parts of plants, and maturity of the plants. In the future, the study design should be standardized to support the reproducibility and sustainability of the data. Since the plants have potential and useful sources of bioactive compounds, further research and evaluation of their safety and efficacy are required. It is also necessary to identify the bioactive substances present in these plants that contribute to their pharmacological activities and their mechanisms of action that are responsible for the pharmacological activities. Finally, further exploration using other *Lepisanthes* species may broaden the discovery of traditional medicine.

## 8. Conclusions

The present review highlighted the botany, traditional uses, phytochemistry, methodological approaches, and the main pharmacological properties of six species from the genus *Lepisanthes*, which may be useful for the formulation of potential candidate drugs. Genus *Lepisanthes* usually grows in a humid climate and has been used traditionally to treat various symptoms including cough, fever, diarrhea, inflammation, and itchiness. Flavonoids and tannins are among the major phenolic classes of *Lepisanthes* species. Both in vitro and in vivo pharmacological experiments, particularly using the ethanolic extracts, have substantially proven the traditional use of *Lepisanthes* species. It demonstrates promising properties against various non-communicable diseases and infectious agents due to its antioxidant, antimicrobial, antihyperglycemic, antidiarrheal, analgesic, and antimalarial properties. For instance, the leaf extracts of *L. rubiginosa* and *L. alata* have shown antihyperglycemic effects, and the compound isolated from the stem and roots of *L. senegalensis* (3-*O*-*trans*-caffeoylbetulinic acid) showed antimalarial activity, while most of the fruits, especially the seed, from this genus showed high antioxidant activity. Nonetheless, most studies were performed as in vitro studies, the safety of *Lepisanthes* is dubious, and the mechanism of action is still unknown. Thus, this warrants a more comprehensive in vivo study to discover the potential bioactive substances from the genus *Lepinsanthes* and their underlying mechanism of action, as it is beneficial for future cost-effective alternative medicine to prevent and treat diseases.

## Figures and Tables

**Figure 1 pharmaceuticals-15-01261-f001:**
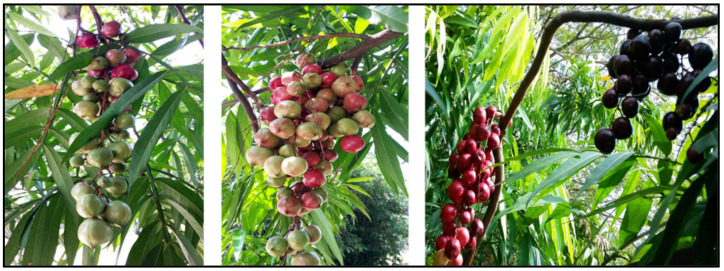
*Lepisanthes alata*.

**Figure 2 pharmaceuticals-15-01261-f002:**
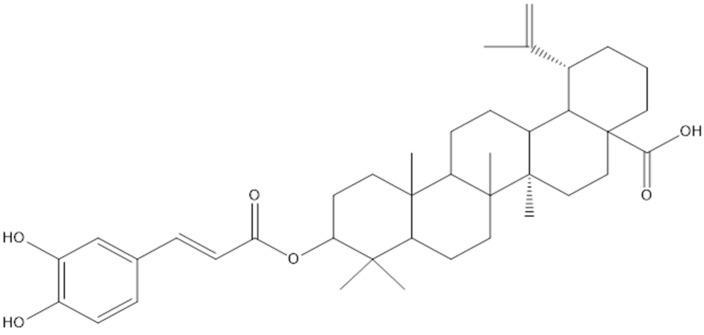
Chemical structure of 3-*O-trans*-caffeoylbetulinic acid.

**Figure 3 pharmaceuticals-15-01261-f003:**
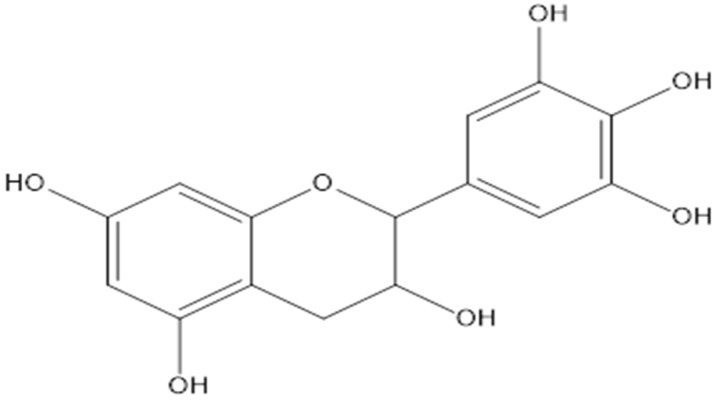
Chemical structure of gallocatechin.

**Figure 4 pharmaceuticals-15-01261-f004:**
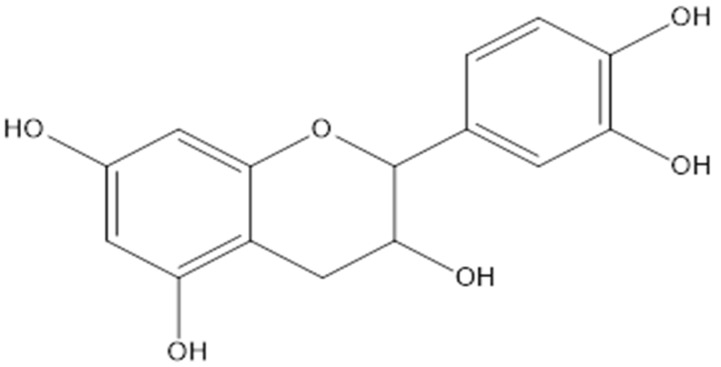
Chemical structure of epicatechin.

**Table 1 pharmaceuticals-15-01261-t001:** Summary of traditional uses of the genus *Lepisanthes*.

Species	Plant Parts	Traditional Medicine Applications	Region/Country
*L. tetraphylla*	Leaves	Cough and fever	Malaysia [12]
	Root	Diarrhea	Bangladesh [13]
	Seed	Dandruff	India [14]
*L. senegalensis*	Root	Malaria, fever with vertigo, chest pain, and nosebleed	Thailand [15]
	Root	Diarrhea	Bangladesh [16]
	Leaves	Bacterial and fungal infections, pain, inflammation, and asthenia	Senegal [17]
*L. fruticosa*	Root	Itchiness and fever	Malaysia [19,20]
	Root	Rheumatism, backache, and to maintain vitality	Sarawak [20]
*L. amoena*	Leaves	Facial skin problems	East Kalimantan [22]
*L. rubiginosa*	Leaves	Muscle soreness	Terengganu [23]
	Fruit	Fever, flatulence, and postpartum blues	Terengganu [23]
	Fruit	Diarrhea, dysentery, and jaundice	Bangladesh [24]
*L. alata*	Leaves	Skin itchiness due to scabs	East Kalimantan [25]

*L.*—*Lepisanthes*.

**Table 2 pharmaceuticals-15-01261-t002:** Phytochemical compounds of *Lepisanthes* species.

Ref	Type	Compound	Species
[19,27]	Flavanol	Gallocatechin	*L. fruticosa*, *L. alata*
		Epicatechin	*L. fruticosa*, *L. alata*
[19]	Flavanonol	Dihydrokaempferol-5-*O-β*-D-glucopyranoside	*L. fruticosa*
		2,5,7-trihydroxyflavanone-4′-*O-β*-D-glucoside	*L. fruticosa*
		Neoastilbin	*L. fruticosa*
	Flavonol	Quercetin-3,7-*O-β*-D-diglucopyranoside	*L. fruticosa*
		Quercetin-3-*O-β*-D-galactopyranoside	*L. fruticosa*
		Kaempferol-3,7-diglucoside	*L. fruticosa*
		Quercetin-3-galactoside-7-glucoside	*L. fruticosa*
		Quercetin-3-*O-β*-D-galactopyranoside	*L. fruticosa*
		Rutin	*L. fruticosa*
		Quercetin-3-sulphate	*L. fruticosa*
		Buddlenoid A	*L. fruticosa*
		Hibiscetin-3-*O*-glucoside	*L. fruticosa*
	Flavone	5,2′-dihydroxy-6,7,8-trimethoxyflavone-2′-*O-β*-D-glucoside	*L. fruticosa*
	Isoflavone	Genistein-7,4′-di-*O-β*-D-glucoside	*L. fruticosa*
	Anthocyanin	Luteolinidin	*L. fruticosa*
[15]	Lupane	28-*O*-acetyl-3 *β-O*-*trans*-caffeoylbetulin	*L. senegalensis*
		3-*O-trans*-caffeoylbetulin	*L. senegalensis*
		3-*O-trans*-caffeoylbetulinic acid	*L. senegalensis*
		Betulin	*L. senegalensis*
		Betulinic acid	*L. senegalensis*
		Lupeol	*L. senegalensis*
		3-*O*-*trans*-caffeoyllupeol	*L. senegalensis*
	Hopane	3α-*O-trans -p*-coumaroyl-22-hydroxyhopane	*L. senegalensis*
		3α-*O-cis*-*p*-coumaroyl-22-hydroxyhopane	*L. senegalensis*
		3α-*O-trans*-caffeoyl-22-hydroxyhopane	*L. senegalensis*
[19]	Other compounds	Mangiferin	*L. fruticosa*
		6-gingerol	*L. fruticosa*
		Ellagic acid	*L. fruticosa*
	Tannin	Procyanidin B2	*L. fruticosa*
		Procyanidin B3	*L. fruticosa*
		Arecatannin A1	*L. fruticosa*
		Arecatannin A2	*L. fruticosa*
		1,2,6-tri-*O*-galloyl*-β*-D-glucopyranoside	*L. fruticosa*
[15]		2,6-dimethoxy-1,4-benzoquinone	*L. senegalensis*

*L.*—*Lepisanthes*.

**Table 3 pharmaceuticals-15-01261-t003:** Overall pharmacological activities of *Lepisanthes species*.

Species	Plant Parts	Extract	Pharmacological Activities (In Vitro)	Pharmacological Activities (In Vivo)
*L. alata* (Blume) Leenh	Fruits (seed, flesh, and peel), leaves, bark	EtOH, MeOH, H_2_O	Antioxidant [5,29], Antimicrobial [5], Antihyperglycemic [27,30] Toxicity [29]	None
*L. amoena* (Hassk.) Leenh	Leaves, stem	EtOH, MeOH, N-hexane, and EtOAc	Antioxidant [21,31], Antimicrobial [31,32]	None
*L. fruticosa* (Roxb.) Leenh	Fruits	Chloroform, hexane, EtOAc, EtOH	Antioxidant [19], Antihyperglycemic [19]	None
*L. senegalensis* (Poir.) Leenh	Leaves, stems, roots	EtOH EtOAc, MeOH	Antimicrobial [17], Antimalarial [15], Toxicity [15,17]	None
*L. rubiginosa* (Roxb.) Leenh	Stem, leaves, bark	EtOH, MeOH	Antioxidant [28], Antimicrobial [33], Antimalarial [34]	Analgesic [28], Antidiarrheal [28], Antihyperglycemic [28]
*L. tetraphylla* (Vahl.) Radlk	Leaves	MeOH, AgNPs	Antimicrobial [35]	None

*L.—Lepisanthes*; EtOH—ethanol; MeOH—methanol; EtOAc—ethyl acetate; AgNPs—silver nanoparticles.

**Table 4 pharmaceuticals-15-01261-t004:** Summary of the methods and antioxidant activities of *Lepisanthes* species.

Ref	Species	Extraction	Part(s) Used	Antioxidant Test	Positive Controls	Findings
[5]	*L. alata* (Blume) Leenh	60% Ethanol	Peel, flesh, seed	DPPH	BHT, vit. E, vit. C	The ethanolic extracts of seed (83.9%) and peel (83.2%) had significantly higher antioxidant activities than the flesh (52.4%) and controls except for vit. C (88.2%).
			Peel, flesh, seed	ABTS	Trolox	The antioxidant activities of seed (48.2%) and peel (45.1%) were still the highest but significantly lower than the Trolox (64.7%).
[29]		Aqueous, methanol, and ethanol	Rind, flesh, seeds, whole fruits, leaves, and bark	DPPH	-	Ethanol and methanol extracts had higher DPPH radical scavenging activities compared to aqueous except for flesh. The ethanolic extracts of bark (93%) and seed (90%) had significantly higher antioxidant activities than the other parts.
[21]	*L. amoena* (Hassk.) Leenh	Ethanol	Flesh, seed, pericarp	DPPH	Vit. C	The ethanolic extracts of the pericarp (IC_50_ 53.21 ppm) and seed (IC_50_ 63.31 ppm) had higher antioxidant activities than the flesh (IC_50_ 122.51 ppm) but were lower compared to vit. C (IC_50_ 3.06 ppm).
[31]		Methanol, 50% ethanol	Stem, leaves	DPPH	Catechin	All the extracts were unable to inhibit the oxidation reaction of DPPH by 50% with catechin used as a positive control.
[19]	*L. fruticosa* (Roxb.) Leenh	Hexane, chloroform, ethyl acetate, and ethanol	Pulp and seed of unripe fruit	DPPH	BHT, vit. C, vit. E	DPPH scavenging activities: Seed extracts: (i) ethanol (IC_50_ 0.178 mg/mL), ethyl acetate (IC_50_ 5.351 mg/mL), chloroform (not detected), hexane (not detected). Pulp extracts: (i) ethanol (IC_50_ 0.207 mg/mL), ethyl acetate (IC_50_ 4.396 mg/mL), chloroform (IC_50_ 13.613 mg/mL), hexane (IC_50_ 29.151 mg/mL). Unripe ethanolic seed extract had stronger scavenging activity (IC_50_ 0.178 mg/mL) than the ethanolic pulp extract (IC_50_ 0.207 mg/mL), BHT (IC_50_ 1.154 mg/mL), vit. C (IC_50_ 0.087 mg/mL), and vit. E (IC_50_ 0.210 mg/mL).
				β-carotene bleaching assays	Vit. C	Β-carotene bleaching (%): Seed extracts: (i) ethanol (70%), ethyl acetate (40%), chloroform (−192%), hexane (−345%). Pulp extracts: (i) ethanol (49.5%), ethyl acetate (50.5%), chloroform (−3.8%), hexane (−290%). Ethanolic seed extract had the highest antioxidant activity (70%) compared to vit. C (25.8%).
[28]	*L. rubiginosa* (Roxb.) Leenh	Ethanol	Leaves	DPPH	Vit. C	Ethanolic leaf extracts had greater antioxidant activity (IC_50_ 31.62 μg/mL) compared to the vit. C (IC_50_ 12.02 μg/mL).

DPPH—2,2-diphenyl-1-picrylhydrazyl; ABTS—2,2-azinobis (3-ethyl benzothiazoline-6-sulfonic acid) diammonium salt; vit.—vitamin; BHT—butylated hydroxytoluene; IC_50_—concentration causing 50% inhibition.

**Table 5 pharmaceuticals-15-01261-t005:** Summary of the methods and antimicrobial activities of genus *Lepisanthes*.

Ref	Species	Part/s Used	Study Design	Model	Extract	Positive Control	Findings
[5]	*L. alata* (Blume) Leenh	Fruits (seed, peel, and flesh)	Bacteria: *B. subtilis*, *B. cereus*, *Listeria monocytogene*, *S. aureus*, *Salmonella* *enterica serovar Typhimurium*, *E. coli*, and *P. aeruginosa*.	In vitro	60% ethanol	Ampicillin, oxytetracycline, and chloramphenicol	The seed extract had the largest zone of inhibition against G +ve bacteria excluding *L. monocytogenes* and G −ve bacteria.
[32]	*L. amoena (Hassk.) Leenh*	Young, semi-mature, and mature leaves	Agar disc diffusion method. Bacteria: *Propionibacterium acnes*, *Streptococcus mutans*. Fungus: *C. albicans*.	In vitro	N-hexane, ethyl acetate, ethanol	-	Ethanolic extract of mature leaves’ antimicrobial activity had the highest inhibition zone against *P. acnes* (12.00 ± 0.00 mm) and *C. albicans* (16.11 ± 0.19 mm).
[31]		Stem and leaves	Antibacterial assay against *Propionibacterium acnes*.		Methanol, 50% ethanol	Chloramphenicol, tetracycline, isopropyl methylphenol	Stem: Methanol (MIC 1.0 mg/mL), 50% ethanol (MIC 1.0 mg/mL), (MBC 2.0 mg/mL). Chloramphenicol: MIC 0.13 mg/mL, MBC 0.13 mg/mL Tetracycline: MIC 0.03 mg/mL, MBC 0.03 mg/mL Isopropyl methylphenol: MIC 1.0 mg/mL, MBC 1.0 mg/mL
[33]	*L. rubiginosa* (Roxb.) Leenh	Bark	Bacteria: *B. cereus, S. aureus*, *Shigella dysenteriae*, *Salmonella typhi.* Fungi: *Aspergillus niger* and *C. albicans*.	In vitro	Methanol	Cephradin, griseofulvin	A significant zone of inhibition was present in both G −ve bacteria but only in one G +ve bacteria (*S. aureus*). G +ve (*B. cereus*), had the maximum resistance. Minimal antifungal activity.
[17]	*L. senegalensis* (Poir.) Leenh	Leaf	Bacteria: *S. aureus*, *Enterococcus faecalis*, *P. aeruginosa* and *E. coli.* Fungi: *A. fumigatus*, *C. neoformans* and *C. albicans*.	In vitro	Ethanol	Gentamicin, amphotericin B	The antibacterial activity of the extract was greater than the positive control against both *S. aureus* and *Enterococcus faecalis*. Low antifungal activity in all three tested fungi.
[35]	*L. tetraphylla* (Vahl.) Radlk	Leaf	Methicillin-resistant *S. aureus* (MRSA), extended-spectrum beta-lactamase-producing *E. coli* (ESBL *E. coli*), multidrug-resistant *P. aeruginosa*, and multidrug-resistant *Acinetobacter* species.	In vitro	Silver nanoparticles (AgNPS) synthesized using aqueous extract and methanol	Amikacin, piperacillin–tazobactam, polymyxin B	AgNPs significantly inhibited bacterial growth against multi-drug resistant *S. aureus*, *E. coli*, *P. aeruginosa*, and *Acinetobacter* species, while crude methanolic leaf extract only inhibited the growth *E. coli* at different concentrations.

*L.*—*Lepisanthes*; *B.*—*Bacillus*; *S.*—S*taphylococcus*; *P. aeruginosa*—*Pseudomonas aeruginosa, E.*—*Escherichia*; *C.*—*Candida*; G +ve—Gram-positive, G −ve—Gram-negative; MIC—minimum inhibitory concentration; MBC—minimum bactericidal concentration (MBC).

**Table 6 pharmaceuticals-15-01261-t006:** Summary of the methods and antihyperglycemic activities of genus *Lepisanthes*.

Ref	Species	Part(s) Used	Study Design	Model	Extract	Positive Control	Effect/Observation
[27]	*L. alata* (Blume) Leenh	Mature leaf	α-amylase and α-glucosidase inhibitory activity	In vitro	Aqueous	Acarbose	Aqueous extracts of mature leaves are 8.5 times more powerful than acarbose in inhibiting α-amylase.
[30]		Young and mature leaf	α-amylase and α-glucosidase inhibitory activity	In vitro	Aqueous and its proanthocyanidis	Acarbose	Young leaves: Aqueous extracts 2.6 times and proanthocyanidins 5.3 times more potent than acarbose. Mature leaves: Aqueous extracts 8.5 times and proanthocyanidins 11.5 times more potent than acarbose. Inhibitory activities against starch hydrolase are age-dependent and increased as the leaves matured.
[19]	*L. fruticosa* (Roxb.) Leenh	Seed, pulp	α-amylase and α-glucosidase inhibitory activity	In vitro	Hexane, chloroform, ethyl acetate, and ethanol	Acarbose	Ethanolic seed crude extracts of *L. fruticosa* had a significant inhibitory effect compared to acarbose, followed by pulp extract against α-glucosidase (*p* < 0.05). The maximum α-amylase inhibitory effect was observed in ethyl acetate seed extract followed by ethanolic pulp and ethyl acetate pulp.
[28]	*L. rubiginosa* (Roxb.) Leenh	Leaf	Oral glucose solutions induced hyperglycemia in Swiss-albino mice	In vivo	Ethanol	+ve control: 5 mg/kg glibenclamide −ve control: 1% Tween-80 and water	Ethanolic extract (250 and 500 mg/kg BW) reduced blood glucose levels in a dose-dependent manner.

*L.*—*Lepisanthes*; BW—body weight.

**Table 7 pharmaceuticals-15-01261-t007:** Summary of the methods and other pharmacological activities of genus *Lepisanthes*.

Ref	Pharmacological Activity	Species	Part (s) Used	Study design	Model	Extract	Positive control	Findings
[28]	Antidiarrheal	*L. rubiginosa* (Roxb.) Leenh	Leaf	Castor oil-induced diarrhea in Swiss-albino mice	In vivo	Ethanol	+ve control: loperamide 3 mg/kg −ve control: 1% Tween-80 and water	Ethanolic extract inhibits acute diarrhea in a dose-dependent manner. Percent of inhibition of defecation: (1) Loperamide group (88.59%) (2) Tested group 500 mg/kg BW extract (77.19%) (3) Tested group 250 mg/kg BW extract (57.89%)
[28]	Analgesic	*L. rubiginosa* (Roxb.) Leenh	Leaf	Acetic acid writhing tests in Swiss-albino mice	In vivo	Ethanol	+ve control: Diclofenac-NA 25 mg/kg −ve control: 1% Tween-80 and water	Ethanolic extract inhibits writhing test in a dose-dependent manner. Percent of inhibition of writhing reflex: (1) Diclofenac-NA group (86.52%) (2) Tested group 500 mg/kg BW extract (58.43%) (3) Tested group 250 mg/kg BW extract (46.07%)
[15]	Antimalarial	*L. senegalensis* (Poir.) Leenh	Stems, roots	Parasites were subcultured in microtitration plates. The indicator of antimalarial activity was by calculating the inhibition of uptake of a radiolabeled nucleic acid precursor by the parasites	In vitro	Ethyl acetate isolated compounds	Mefloquine and dihydroartemisinin	Bioactive compound no. 6 (3-*O*-*trans*-caffeoylbetulinic) exhibited moderate antimalarial activity against *P. falciparum* (IC_50_ 4.5 µM)
[34]		*L. rubiginosa* (Roxb.) Leenh	Stems	Measurement of HRP2 antimalarial assay was performed using ELISA against the *P. falciparum* 3D7 strain (chloroquine-sensitive)	In vitro	80% Ethanol		Stem extract had the highest inhibition value (92.4%) at a concentration of 1000 µg/mL, but its IC_50_ value (IC_50_ 252 µg/mL) was categorized as inactive.

*L.*—*Lepisanthes*; *P. falciparum*—*Plasmodium falciparum*; BW—body weight; IC_50_—concentration causing 50% inhibition.

## Data Availability

Data sharing not applicable.

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
