# Peer review of "Genus Lepisanthes: Unravelling Its Botany, Traditional Uses, Phytochemistry, and Pharmacological Properties"

_pharmaceuticals, 2022, doi:10.3390/ph15101261_

Round 1

Reviewer 1 Report

Minor corrections

1) +ve should be changed by positive, -ve should be changed by negative.

Major corrections

1) It is not clear why a review about the Lepisanthes genus should be published. What is the importance of this genus?

2) An ecological section of this genus should be included.

3) The toxicity of this genus should be included in a section.

4) Section 4. The authors only describe the results. A discussion of this section should be included. What is the most promising pharmacological activity of this genus?

5) A section about future perspectives on this genus should be included.

6) A section about the chemical components (phytochemistry) identified in this genus should be included. The chemical structures of the most promising compounds should be shown. In addition, the description of the pharmacological and toxicological effects of these compounds will improve this review manuscript.

7) Conclusion section should be improved. If the authors include the additional sections, the conclusions could be supported with new information.

Author Response

1

It is not clear why a review about the Lepisanthes genus should be published. What is the importance of this genus?

Lepisanthes genus is widely used in traditional medicine and ongoing in vitro studies had proven its pharmacological properties.  However, the summary of the available evidence remains unknown and is mostly performed in vitro. This review will be very useful for future researchers to understand the common methods and overall pharmacological properties of the Lepisanthes genus before translating it to in vivo and human studies. We have rephrased the abstract.  Please refer line 12-26.

2

An ecological section of this genus should be included.

We have added  the botany section, please refer to line 65-86.

3

The toxicity of this genus should be included in a section.

As suggested, we have shifted this info new section 5. Please refer to line 504-533.

4

Section 4. The authors only describe the results. A discussion of this section should be included. What is the most promising pharmacological activity of this genus?

Section 4 is the overall summary of the pharmacological activities. Further results and discussion can be found in the following subheadings (i.e 4.1,4.2 and etc). As suggested, we have stated the antioxidant activity as the most promising pharmacological activity of this genus. Please refer to line 184-186.

5

A section about future perspectives on this genus should be included.

As suggested, we have included this section. Please refer line 534-548.

6

A section about the chemical components (phytochemistry) identified in this genus should be included. The chemical structures of the most promising compounds should be shown. In addition, the description of the pharmacological and toxicological effects of these compounds will improve this review manuscript.

As suggested, we have included the chemical components (phytochemistry) identified in this genus.  Please refer line 125-176.

7

Conclusion section should be improved. If the authors include the additional sections, the conclusions could be supported with new information.

We have rephrased our conclusion. Please refer line 549-548.

8

It could be interesting to add some pictures of Lepisanthes species (especially the reported ones).

Pictures have been added.  Please refer line 87-89.

9

 It could be interesting to give the list of all 26 species existing in this genus.

In the present study, we focused on the six species only. Therefore, we might not be able to list all 26 species.

10

Please verify the italic of some words (especially the word genus after Lepisanthes).

We have verified the italic of the words following Lepisanthes. It should be written as L. rubiginosa (Roxb.) Leenh.

11

Some abbreviations are not explained (GAE, IC50, G +ve …)

Abbreviations have been explained.

12

 Please correct section number:

4.5. Analgesic Activities

4.6. Antimalarial Activities

5. Conclusion

Corrections have been done.

13

For tables, try to avoid repeating the same sentences already in the text. The columns Study design and Findings or Effect/ observation should contain brief and clear information.

We have rephrased the sentences.

14

Also, it’s better to put antimicrobial activity in a separate table (like for antioxidant in table 2) since it contains several works. That will also allow the table to be closer to the corresponding section.

Table 3 has been separated.

15

Also, please cite table 2 at the beginning of the section (not the end) to allow the readers to follow in the table. Same for table 3, it should be cited at the beginning of every section (Antimicrobial Activities and Toxicity Test > Antimalarial Activities) not only at the end of section “Antimalarial Activities”.

References have been shifted to the first column.

16

Please verify the references: Add Jun et al. (2003) (Line 168) in the list.

Jun et al. (2003) has been added in the list. (√)

Reviewer 2 Report

This review on genus Lepisanthes highlights the traditional use and pharmacological properties of six species Lepisanthes alata, L. amoena, L. fruticosa, L. senegalensis, L. rubiginosa, and L. tetraphylla. Interesting antioxidant and antimicrobial activities have been reported. Some works also investigated the antihyperglycemic, antidiarrheal, antimalarial, and analgesic activities. This review is well written and gathers nicely the investigations published on the genus species.

General comments:

-          It could be interesting to add some pictures of Lepisanthes species (especially the reported ones).

-          It could be interesting to give the list of all 26 species existing in this genus.

-          Please verify the italic of some words (especially the word genus after Lepisanthes).

-          Some abbreviations are not explained (GAE, IC50, G +ve …)

-          Please correct section number:

4.5. Analgesic Activities

4.6. Antimalarial Activities

5. Conclusion

-          For tables, try to avoid repeating the same sentences already in the text. The columns Study design and Findings or Effect/ observation should contain brief and clear information.

Also, it’s better to put antimicrobial activity in a separate table (like for antioxidant in table 2) since it contains several works. That will also allow the table to be closer to the corresponding section.

Also, please cite table 2 at the beginning of the section (not the end) to allow the readers to follow in the table. Same for table 3, it should be cited at the beginning of every section (Antimicrobial Activities and Toxicity Test > Antimalarial Activities) not only at the end of section “Antimalarial Activities”.

-          Please verify the references: Add Jun et al. (2003) (Line 168) in the list.

Specific comments:

L78 : certain species (or plants) are consumed

L 80: the type of plant (species) > not genus since you talk about one genus

L 89-91: please add the country if possible

L 111: please correct the reference Hassan et al. (2017) > Hasan et al. (2017)

Table 1:

-          For L. alata, in column pharmacological activities (in vitro): you have only the study 28 for toxicity (eliminate 15)

-          For L. amoena, in column extract: please add n-hexane and ethyl acetate (cited in antimicrobial reference 30)

-          For L. senegalensis, in column pharmacological activities (in vitro): please add the reference for toxicity (15).

-          For L. rubiginosa, in column plant parts: please add bark, and in column extract: methanol (cited in antimicrobial reference 31).

Section: Antioxidant Activities

-          You have cited reference 29 for L. amoena in table 1 but you have not added their work in this section.

-          L 118-127: since you only have 3 tests, I suggest you add β-carotene bleaching assay in this paragraph

-          L 131-132: I suggest you add some details on the work of Anggraini et al. (2019) [25]. Since you have only few investigations, it could be interesting to discuss this work too.

-          L 143: Please correct the name of the author in reference Mirfat et al. (2020); it is Salahuddin. You have 2 references for this author. You cite it in table 1 as 17 (Salahuddin et al., 2020) and in table 2 as 36 (Salahuddin et al., 2017). Please correct its number in table 2 (17) and add it at the end of sentences L 145 and 149 to avoid confusion with reference 36. You can develop the result of study 36, cited in L145-147, to enrich the manuscript (if you do so, add it in table 1 too).

-          In Table 2: please specify in column findings for L. fruticose, test DPPH, that you have cited the results of ethanolic extracts since you have several extracts. Also, add the results of the other extracts if possible. In β-carotene bleaching assay, please add the results in the table (%).

-          L 163: explain the abbreviation GAE (Gallic Acid Equivalent).

-          L 164: Please cite the number of the reference of Salusu et al. (2017) at the end of the first sentence. You have cited Jun et al. (2003) before giving the number of Salusu, which causes confusion.

-          L 172-174: please add IC50 value of vit C in the table.

-          L 182: Please add the meaning of IC50.

Section Antimicrobial Activity and Toxicity Test

-          L 188-189: to clarify the work, please specify which strains are bacteria and which are fungi.

-          L 189-190, 193-195: The solvent ethanol was not mentioned before, you have only cited n-hexane, ethyl acetate, and methanol extracts. Please verify it in the table too.

-          For the specie L. amoena, you have also cited reference 29 in table 1. Please discuss it here and add it to the table.

-          L 212: L. Senegalensis > L. senegalensis

-          L 218-219: please add the antifungal activity in the table, column study design.

-          L 236-247: please add the number of the reference Anggraini et al. (2019) (28)

-          L 244-246: please add the number of the reference Dior et al. (2017) (15).

-          In the table: it’s better to add the methods in column study design. For Effect/observation column, clear results (values) might be better than sentences. Also, explain the abbreviation G +ve and G -ve.

Section: Antihyperglycemic Activities

-          For L. alata, you have cited reference 26 in table 1. Please add it here.

-          L 298-308: please add the solvent (ethanol) in table 3 for the corresponding work.

Section : Antimalarial Activities

-          L346-355 : since this work was done on isolated compounds and not on the extract, it’s better to add in table 3, column extract: Ethyl acetate isolated compounds.

Table 3

-          L 386: add P. falciparum abbreviation

Conclusion

-          L393: it’s not the stem and roots but the compound N°6 (3-O-trans-caffeoylbetulinic acid) that showed antimalarial activity. Please specify it.

Author Response

1.

L78: certain species (or plants) are consumed

The word Lepisanthes genus have already been changed to species in L:96

2.

L 80: the type of plant (species) > not genus since you talk about one genus

The word genus has been changed to species in L:98

3.

L 89-91: please add the country if possible

The country is already added in L: 105

4.

L 111: please correct the reference Hassan et al. (2017) > Hasan et al. (2017)

Correction has been done in L: 182

5

Table 1:

For L. alata, in column pharmacological activities (in vitro): you have only the study 28 for toxicity (eliminate 15)

Reference 15 already eliminated in column pharmacological activities (in vitro): the study for toxicity. Currently in table 3.

 For L. amoena, in column extract: please add n-hexane and ethyl acetate (cited in antimicrobial reference 30)

N-hexane and ethyl acetate already added in column extract for L. amoena in Table 3. Summary of the methods and antimicrobial activities of genus Lepisanthes.

For L. senegalensis, in column pharmacological activities (in vitro): please add the reference for toxicity (15).

The reference for toxicity (15) already added in Table 3, in column pharmacological activities (in vitro): for L. senegalensis.

For L. rubiginosa, in column plant parts: please add bark, and in column extract: methanol (cited in antimicrobial reference 31).

Bark added in column plant parts and methanol added in column extract in table 3.

6.

Section: Antioxidant Activities

You have cited reference 29 for L. amoena in table 1 but you have not added their work in this section.

The elaboration of the study already added in line 262-269.

L 118-127: since you only have 3 tests, I suggest you add β-carotene bleaching assay in this paragraph

β-carotene bleaching assay addeed in line 202-206 .

L 131-132: I suggest you add some details on the work of Anggraini et al. (2019) [25]. Since you have only few investigations, it could be interesting to discuss this work too.

Details regarding Anggreini et al. studies added in line 221-227.

L 143: Please correct the name of the author in reference Mirfat et al. (2020); it is Salahuddin. You have 2 references for this author. You cite it in table 1 as 17 (Salahuddin et al., 2020) and in table 2 as 36 (Salahuddin et al., 2017). Please correct its number in table 2 (17) and add it at the end of sentences L 145 and 149 to avoid confusion with reference 36. You can develop the result of study 36, cited in L145-147, to enrich the manuscript (if you do so, add it in table 1 too).

The name of author corrected in L228.

Table 2 alredy corrected to reference (17).

Reference (17) already added to the end of sentence in L: 237.

In Table 2: please specify in column findings for L. fruticose, test DPPH, that you have cited the results of ethanolic extracts since you have several extracts. Also, add the results of the other extracts if possible. In β-carotene bleaching assay, please add the results in the table (%).

Results form other solvent used added in table 4.

Results for B-carotene bleaching assays added in table 4.

L 163: explain the abbreviation GAE (Gallic Acid Equivalent).

The abbreviation for GAE (Gallic Acid Equivalent) added in L: 251

L 164: Please cite the number of the reference of Salusu et al. (2017) at the end of the first sentence. You have cited Jun et al. (2003) before giving the number of -Salusu, which causes confusion.

The citation for Salusu et. al already added in L: 262

L 172-174: please add IC50 value of vit C in the table.

Vit C IC50 value added in table 4.

L 182: Please add the meaning of IC50.

Meaning for IC50 stated in L:274.

7.

Section Antimicrobial Activity and Toxicity Test

-  L 188-189: to clarify the work, please specify which strains are bacteria and which are fungi.

Both strains are clarified in L:286

-   L 189-190, 193-195: The solvent ethanol was not mentioned before, you have only cited n-hexane, ethyl acetate, and methanol extracts. Please verify it in the table too.

The solvent used in this study are n-hexane, ethyl acetate and ethanol not methanol. Correction has been made L283 as well as in Table 5.

-          For the specie L. amoena, you have also cited reference 29 in table 1. Please discuss it here and add it to the table.

The antimicrobial study of referrence 29/ 31 already added in L: 296-302 and also added in table 5.

-          L 212: L. Senegalensis > L. senegalensis

Correction made in L 316

-          L 218-219: please add the antifungal activity in the table, column study design.

The antifungal activity added in the colum study design.

-          L 236-247: please add the number of the reference Anggraini et al. (2019) (28)

This paragraph already placed under 5. Toxicity

-          L 244-246: please add the number of the reference Dior et al. (2017) (15).

This sentences are deleted from the paragraph.

-          In the table: it’s better to add the methods in column study design. For Effect/observation column, clear results (values) might be better than sentences. Also, explain the abbreviation G +ve and G -ve.

Some clear values are added and some are maintained as the results are completely clear. The abbrevation for Gram-positive and Grams-negative added.

8

Section: Antihyperglycemic Activities

-          For L. alata, you have cited reference 26 in table 1. Please add it here.

-          L 298-308: please add the solvent (ethanol) in table 3 for the corresponding work.

The reference 26 added in the table as well as in the paragraph.

Solvent Ethanol added in table 6.

Section : Antimalarial Activities

-          L346-355 : since this work was done on isolated compounds and not on the extract, it’s better to add in table 3, column extract: Ethyl acetate isolated compounds.

Table 3

-          L 386: add P. falciparum abbreviation

Ethyl acetate isolated compounds added in table 7 colum extract.

Plasmodium falciparum abbreviation added.

Conclusion

-          L393: it’s not the stem and roots but the compound N°6 (3-O-trans-caffeoylbetulinic acid) that showed antimalarial activity. Please specify it.

The name of the compound included in L: 562

Round 2

Reviewer 1 Report

The manuscript can now be accepted for its publication in Pharmaceuticals